# Mutual Enhancement of Opioid and Adrenergic Receptors by Combinations of Opioids and Adrenergic Ligands Is Reflected in Molecular Complementarity of Ligands: Drug Development Possibilities

**DOI:** 10.3390/ijms20174137

**Published:** 2019-08-24

**Authors:** Robert Root-Bernstein, Beth Churchill, Miah Turke, Udaya K. Tiruttani Subhramanyam, Joerg Labahn

**Affiliations:** 1Department of Physiology, 567 Wilson Road, Room 2201 Biomedical and Physical Sciences Building, Michigan State University, East Lansing, MI 48824, USA; 2Centre for Structural Systems Biology (CSSB), Notkestraße 85, 22607 Hamburg, Germany; 3Forschungszentrum Juelich GmbH, ICS-6, 52425 Juelich, Germany

**Keywords:** coevolution, interactome, opiate receptors, adrenergic receptors, opioids, enhancement, tachyphylaxis, fade, prevention, reversal, drug development, combination drugs, linked compounds, tethered drugs

## Abstract

Crosstalk between opioid and adrenergic receptors is well characterized and due to interactions between second messenger systems, formation of receptor heterodimers, and extracellular allosteric binding regions. Both classes of receptors bind both sets of ligands. We propose here that receptor crosstalk may be mirrored in ligand complementarity. We demonstrate that opioids bind to adrenergic compounds with micromolar affinities. Additionally, adrenergic compounds bind with micromolar affinities to extracellular loops of opioid receptors while opioids bind to extracellular loops of adrenergic receptors. Thus, each compound type can bind to the complementary receptor, enhancing the activity of the other compound type through an allosteric mechanism. Screening for ligand complementarity may permit the identification of other mutually-enhancing sets of compounds as well as the design of novel combination drugs or tethered compounds with improved duration and specificity of action.

## 1. Introduction

Many receptor systems engage in crosstalk, modifying each other’s activity. Understanding how such crosstalk evolved and the molecular mechanisms that mediate receptor interactions would obviously be a boon to drug developers, especially if, in addition, it were possible to predict such interactions in advance and to take advantage of their unique properties. We have been working on these problems for almost two decades and have generated a set of simple methods that seem to have some predictive value. We illustrate the application of these methods here with reference to understanding crosstalk between the opioid and adrenergic receptor systems. We believe that the results may have value for the development of novel, enhanced opioid and adrenergic drugs.

Interactions between adrenergic and opioid drugs are well studied. Opioid peptides and catecholamines are co-stored and co-released in neurons and the adrenals [1,2,3,4,5]. Alpha 2A-adrenergic receptors (A2ADRs) and mu opioid receptors (muOPRs) are co-localized within various neurons or are expressed on adjoining neurons that share synapses [6,7]. ADR and OPR co-localize and dimerize in the membranes of cells expressing both types of receptors [8,9,10,11,12,13,14,15,16]. Co-functionality is such that ADR controls the locomotor and reward effects of opioids [17,18]. Opioids (including both enkephalins and morphine) as well as opioid antagonists (e.g., naloxone and naltrexone) enhance adrenergic drug activity [19,20,21,22,23,24,25,26,27,28,29,30,31,32,33] via an extracellular receptor-mediated mechanism [34,35,36,37,38,39]. Conversely, adrenergic drugs (particularly epinephrine, clonidine, and amphetamines) enhance opioid receptor activity [40,41,42,43,44,45,46,47,48,49,50,51,52,53,54,55], again by means of an extracellular binding site [56]. In both cases, the enhancement is such that in the presence of the enhancer, the primary ligand attains full activity at significantly lower concentrations than when the ligand is alone. The duration of the activity produced by the combination is significantly longer than when the primary ligand is delivered alone. Perhaps most exciting from a pharmaceutical perspective, epinephrine, clonidine, and other adrenergic agonists inhibit the development of tachyphylaxis caused by the down-regulation of receptors in opiate analgesia [57,58,59] and can reverse “acute tolerance” (i.e., fade or tachyphylaxis) caused by repeated doses of morphine on guinea pig ileum [60,61,62]. Conversely, opioids inhibit the development of tachyphylaxis and fade caused by the down-regulation of adrenergic receptors [32,63]. Most surprisingly, adrenergic compounds have been demonstrated to bind to low affinity opioid binding sites on opioid receptors [56,64], while opioids bind to a site immediately adjacent to the high affinity binding site of adrenergic receptors [39,40,41,65,66].

In previous papers, we proposed some general hypotheses regarding the evolutionary principles governing the emergence of integrated molecular systems (interactomes). We test some of the implications of these hypotheses here with regard to understanding how opioid and adrenergic receptors co-evolved to “talk” to one another. Our general hypotheses are as follows:(1)Evolution builds on the reversible interactions of molecular complementary structures that balance specific binding to protect participating molecules against degradative processes while creating transient complexes with novel functions [67,68,69].(2)In consequence, every molecule involved in living systems interacts more or less specifically, strongly, and transiently with several others to form the chemical basis of an “interactome” [70].(3)Molecular complementarity may be expressed either as homocomplementarity (self-aggregation) or heterocomplementarity (binding of two, unlike ligands) [71,72,73,74,75,76].(4)Ligands and their receptors and transporters evolved from molecularly complementary structures that become more elaborated and specialized over time, thereby forming functional modules with (identifiable) conserved structural elements [71,72,73,74,75,76].(5)Receptors for molecularly complementary ligands evolve complementary functions. For example, a homocomplementary molecule may evolve into a ligand–receptor pair, or heterocomplementary molecules may evolve so that one becomes the ligand and the other the receptor.(6)As a result of principles 1–5, compounds that are molecularly complementary will alter each other’s physiological activity, and conversely, when compounds alter each other’s physiological activity, they will be found to be molecularly complementary [77].(7)Finally, there is a molecular paleontology within modern proteins that provides clues to how complementary modules were selected for and employed in any given set of receptors and transporters [70,74].

We previously used these principles to discover and establish the fact that adrenergic and opioid receptors share common ligand binding, though in different regions of their receptors. Adrenergic receptors have opioid binding sites consisting of their first two extracellular loops [34,35,36,37,38,39] while the mu opioid receptor has an adrenergic binding site consisting of its first two extracellular loops [56,64]. In each case, the binding of these enhancers alters the binding affinity of the primary ligand, shifting the dose response curve to the left by almost a log unit and increasing the duration of receptor activation [56,65,77]. Thus, our previous work has established a mechanistic basis for the receptor-mediated integration of opioid–adrenergic interactions. 

In light of the documented interactions between the opioid and adrenergic systems, we test principles (1)–(4) to determine whether there is a stereochemically specific molecular complementarity between adrenergic and opioid ligands that corresponds to receptor complementarity. We draw out possible drug design implications of such complementarity. 

## 2. Results

### 2.1. Opioid Compounds Bind to Adrenergic Compounds

Because opioids and adrenergic compounds modify each other’s physiological activities, these sets of compounds should be molecularly complementary and thus bind specifically to each other according to our first proposition. This appears to be the case (Table 1 and Figure 1, Figure 2 and Figure 3).

In general, compounds such as epinephrine, norepinephrine, and dopamine bind to opioid compounds such as methionine-enkephalin, morphine, and even the opioid antagonist naloxone, with binding constants in the high nanomolar to low micromolar range. In contrast, the amino acid precursors of the adrenergic compounds, tyrosine and phenylalanine, have much higher binding constants (meaning that much greater concentrations are necessary to produce the same effect). The binding seems to be specific for adrenergic compounds, since histamine and acetylcholine have little binding, but serotonin also exhibits high affinity for opioids. Glucose and riboflavin generally exhibit little binding to any of the classes of compounds but ascorbic acid binds with moderate affinity for many adrenergic compounds, though not to serotonin, histamine, acetylcholine, or the opioids (the latter result not shown in the table). Notably, ascorbic acid enhances the physiological activity of adrenergic compounds [39,65]. Some of these binding results were previously demonstrated using nuclear magnetic resonance spectroscopy [78] and capillary electrophoresis [79] (see Table 1).

### 2.2. Adrenergic Compounds Bind to Opioid Receptor Peptides

A second prediction that follows from our principles of receptor co-evolution is that receptors for each of a pair of molecularly complementary compounds should evolve to bind the complementary compound, using it as an allosteric modulator of function. In this section, we report our experiments testing the first part of this prediction regarding receptor binding. UV spectroscopy was used to test for the binding of a range of aminergic and opioid compounds (and controls such as acetylcholine) to peptides derived from extracellular and transmembrane regions of the mu opioid receptor (muOPR) and various aminergic receptors (dopamine, histamine, and beta adrenergic). Some experimental results are shown in Figure 4, Figure 5 and Figure 6 and a broader set is summarized in Table 2. 

As Table 2 reveals, opioid compounds more often bind to the extracellular loop regions of aminergic receptors with higher affinities than they do to extracellular regions of opioid receptor peptides. Conversely, aminergic compounds more often bind to the extracellular regions of the opioid receptor peptides than they do to aminergic receptor loops Note that, in general, adrenergic compounds bind to some regions of both the muOPR and the aminergic receptors, whereas (with one exception) histamine and acetylcholine do not bind to any of the receptor peptides tested. The one exception is that histamine binds to the H1HR 77–87 peptide, which is from an extracellular loop, suggesting that this loop may help to guide histamine into the receptor pocket. The binding of morphine (Mor), naloxone (Nalox), and methionine-enkephalin (M-Enk) to aminergic receptors has previously been associated with the allosteric enhancement of amine receptor function, while aminergic binding to muOPR has previously been associated with the allosteric enhancement of opiate receptor function (see Discussion below). Note that, in general, epinephrine (Epi), norepinephrine (NorEpi), dopamine (Dop), and amphetamine (Amph) bind better to muOPR extracellular loop peptides than to the aminergic extracellular loop peptides tested here. Acetylcholine (ACh) and histamine (Hist) generally do not bind measurably to any of the receptor peptides tested, with the one exception of histamine binding to one of the H1HR extracellular loops. The binding of the compounds tested to insulin receptor peptides (INSR) and to the transmembrane peptide from the muOPR (121–131) was uniformly two or more orders of magnitude less than to the other receptor peptides. These observations suggest that each set of receptors has evolved to recognize and bind the complementary set of compounds. Acetylcholine and histamine do not, however, display measurable binding to opioid receptor peptides, nor does either compound bind to peptides from the extracellular loops of the other aminergic receptors tested. Thus, the observed binding is limited to the opioid–amine complementarities observed in the bioactive compound binding study (Table 1). 

Notably, some of the ligands display dual binding profiles with high and low affinity binding. These dual affinity binding curves appear to be limited to adrenergic compounds binding to opioid receptor peptides (Figure 4, Figure 5 and Figure 6) and opioid compounds binding to adrenergic receptor peptides (Table 2). These data suggest that the extracellular loop regions of these receptors have evolved to be optimized for the binding of the complementary ligand. 

### 2.3. Epinephrine Binds to Intact Mu Opioid Receptor

The next prediction that we tested involved the binding of adrenergic and opioid compounds to intact mu opioid receptors (muOPRs), which follows logically from their binding to extracellular receptor peptide sequences. We have previously demonstrated that adrenergic compounds, morphine and methionine-enkephalin (Met-Enk), bind to intact mu opioid receptors (muOPRs). Additionally, in the presence of both ligands, the binding of morphine to its receptor is increased 10-fold (a log shift in the binding constant to the left) [56]. These data are consistent with experiments previously reported in Figure 3 of Jordan et al. [10] and by Ventura, et al. [80]. We also previously published data showing that histamine does not bind to intact muOPRs [56]. Here we provide additional evidence that serial additions of increasing concentrations of acetylcholine (Figure 7), like serial additions of buffer or histamine, produce insignificant changes in the muOPR spectrum, but that epinephrine (Figure 8), naloxone (Figure 9), and methadone (Figure 10) each produce very distinctive and different, concentration-dependent alterations in the muOPR spectrum. 

The data were analyzed at both 200 nm and 210 nm. The advantage of analyzing the data at 210 nm is that there is no shift in the spectrum at that wavelength due to solvent dilution during the serial additions of ligand (see Figure 7 and [56]) so that no correction is needed for such effects in calculating binding constants (Table 3). The resulting binding curves (Figure 11) are in the 15 µM to 30 μM range (Table 3). The 200 nm data need to be corrected for concentration-dependent solvent shifts, which make the resulting data less reliable. The 200 nm data, however, reveal higher affinity (lower Kd value) binding in the nanomolar range (Table 3) and the existence of both high affinity and low affinity binding (see also [56]).

Note that these experiments are carried out with a purified receptor that is not incorporated into the cellular membrane. It is probable that the receptor is not, therefore, in its native conformation and thus the binding constants probably do not reflect what would be found in in vivo studies. However, as noted above, our results are consistent with previous reports of adrenergic binding to opioid receptors that were carried out in intact cells [64,65,66]. Our reasons for the use of an isolated, pure receptor were three-fold. First, it has proven impossible to find a cellular membrane that does not have adrenergic receptors, the existence of which would confound adrenergic binding calculations, particularly as we had to presume that adrenergic receptors have higher affinity for adrenergic ligands than do opioid receptors. Secondly, adrenergic receptors are known to modify muOPR affinity for both opioid and adrenergic ligands through dimerization with the OPR. And finally, multiple mechanisms of adrenergic crosstalk with opioid receptors are known (see Introduction) that we could not reasonably control for. Given these difficulties, we chose to use a purified, unincorporated receptor.

What is important in the context of drug design is that the results reported here confirm the aminergic–muOPR peptide binding studies summarized in Table 2 above. Thus, ligand–receptor peptide binding appears to be a good predictor of intact purified receptor binding, which is in turn a reasonable substitute for in vivo binding studies [64,65,66]. We also note that the binding of opioid compounds to extracellular loops of adrenergic receptors also accurately predicted opioid binding to, and enhancement of, intact adrenergic receptors in vivo [19,20,21,39,40,41].

## 3. Discussion

### 3.1. Summary of Observations

To summarize, we demonstrated here that adrenergic and opioid compounds are, as their complementary physiological activities predict, molecularly complementary, binding to each other with high nanomolar to low micromolar affinities (Table 1 and Figure 1, Figure 2 and Figure 3). In particular, adrenergic compounds bind opioids in general, but not various control compounds such as histamine, acetylcholine, and glucose that are often co-existent with opioid receptors. The molecular complementarity of adrenergic and opioid compounds extends to regions of their receptors (Table 2 and Figure 4, Figure 5 and Figure 6). Adrenergic compounds bind with micromolar affinities to extracellular loop peptides derived from the mu opioid receptor while opioid compounds bind with micromolar affinities to extracellular loop peptides derived from adrenergic receptors. The lower affinity binding of each class of compounds to its own extracellular receptor peptides was also observed (Table 2). The complementarity between opioids and adrenergic compounds for each other appears to extend to binding to each other’s receptors (Table 3). Experimental results confirm that the extracellular loops are where the respective complementary enhancers bind and that co-binding results in enhanced receptor activity for both receptor classes [56,64,65]. Ligand complementarity therefore extends to receptor complementarity in the adrenergic–opioid case.

### 3.2. Ligand Complementarity and Drug Development

The molecular complementarity of opioid–adrenergic ligands and the extension of that complementarity to their receptors helps to explain a phenomenon that has plagued drug development since the discovery of these receptor types, which is the often promiscuous binding of aminergic drugs to opioid receptors [64,65,66] and of opioid drugs to aminergic receptors [19,20,21]. On the other hand, the fact that opioids bind to ADR and adrenergic drugs bind to OPR may be viewed as a potential boon to drug development. 

First, ADR–OPR synergism may be manipulated by means of specific combinations of adrenergic and opioid compounds. The normal function of ADR and OPR (illustrated in Figure 12) involves the binding of the ligand to the receptor, which activates, initiating G-protein coupling (Gαβγ) to the intracellular loops of the receptor. Receptor activation is followed a short time later by the release of the ligand, phosphorylation (P) of the receptor by receptor kinases (GRK), and receptor inactivation and internalization. In the presence of combinations of adrenergic and opioid compounds (which, not incidentally, bind to each other, stabilizing and increasing their availability), both sets of receptors are activated simultaneously.

Several important novel effects follow from the co-activation of both ADR and OPR (Figure 12). One is that the binding of the ligand is stabilized by the presence of its complementary enhancer, resulting in the receptor being kept in its high-affinity, activated state for a longer period of time [56,65]. In addition, the concentration of ligand required to produce any given amount of activation is significantly decreased [56,65]. The duration of the activation is also significantly increased, preventing fade and tachyphylaxis [32,60,61,62,63]. The mechanism of the increased duration of activity appears to be interference with the phosphorylation of the receptor normally carried out by GRK [63,65]. Additionally, the presence of both adrenergic and opioid compounds results in the formation of adrenergic–opioid receptor heterodimers [8,9,10,11,12,13,14,15,16] that further modulate the systemic effects of the pair of compounds (Figure 12, Top Left). 

So the first potential drug development opportunity is to optimize the various ways in which adrenergic–opioid combinations can be utilized to activate, or inactivate, specific receptor subtypes. Combinations of alpha agonists with naloxone, for example, would enhance alpha adrenergic receptor activity while enhancing opioid receptor inactivation. Alternatively, a combination of a beta adrenergic antagonist with morphine might enhance beta adrenergic inactivation while simultaneously enhancing opioid receptor activation. Different classes of opioids would similarly activate (or antagonize) particular opioid receptor subtypes (Figure 13, Top Left). 

The second potential drug development opportunity is to develop compounds that antagonize receptor heterodimer formation (Figure 13, Top Right). Since the sequences and structures of the transmembrane (TM) regions involved in both homo- and heterodimer formation are known, drugs that prevent one or the other (or foster such formation) might provide important means of modulating the interactions between opioid and adrenergic functions. 

The third potential drug development opportunity consists of synthesizing tethered compounds that link an adrenergic ligand to an opioid enhancer so as to activate (or antagonize) specifically a particular class of adrenergic receptor (Figure 13, Bottom Left). As with the drug combinations, an adrenergic agonist or antagonist could be chosen that targets a particular adrenergic receptor class and the opioid enhancer could be chosen so as to partially activate opioid receptors (by using an agonist) or to prevent opioid receptor activation (by using an antagonist). 

Finally, the fourth potential drug development opportunity consists of synthesizing tethered compounds that link an opioid ligand to an adrenergic enhancer so as to activate (or antagonize) specifically a particular class of opioid receptor (Figure 13, Bottom Right). Again, as with the drug combinations, an opioid agonist or antagonist could be chosen that targets a particular opioid receptor class and the adrenergic enhancer could be chosen so as to partially activate adrenergic receptors (by using an agonist) or to prevent adrenergic receptor activation (by using an antagonist).

### 3.3. Proofs of Concept for Complementary Ligand Approach to Enhancer Drug Development

We have tested proof-of-concepts for both the general and specific approaches to the drug development opportunities just outlined. Generally, we have outlined and tested a general method for identifying complementary compounds likely to modify each other’s physiological activity [40,41,77]. The general method begins by identifying molecularly complementary compounds. This identification can either begin with experiments to find a molecular complement to a drug whose activity one desires. In a second step, the potential enhancing compounds are tested for binding to the drug receptor to ascertain their potential to act as allosteric modifiers of the receptor activity. The third step is then to screen the remaining set of potential enhancers for enhanced (or inhibited) activity in vitro or in vivo in the presence of the drug or ligand of interest. We have validated this method by using it to discover a novel class of ADR enhancers, involving tartaric acids based on the fact that tartaric acids have the same binding motif as ascorbic acid and bind to adrenergic agonists [41]. 

We have also tested the feasibility of developing tethered drugs of the sort just suggested. We discovered that, like opioids, ascorbic acid also enhances ADR [39,40,65], and synthesized a compound consisting of ascorbic acid tethered by a four-unit polyethylene linker to epinephrine [81]. This tethered compound had slightly lower affinity for the ADR than epinephrine but retained its enhanced activity in terms of dosage and duration of action. Thus, it is possible that screening for ligand complementarity may provide a simple means of identifying drugs with the potential specifically to enhance or inhibit each other’s activity. 

This small-molecule complementarity approach appears to be a generalizable method for identifying potentially useful drug interactions. For example, it is well known that flavins such as riboflavin bind to serotonin-like molecules [82,83,84,85] and that flavins interact physiologically with psychotropic drugs of the serotonin class [86]. Oddly, then, while both riboflavin and serotonin antagonists are used to treat migraines [87,88], there appears to be no research on the use of the two together. Root-Bernstein and Dillon [77] have reviewed many other cases of small molecule complementarity that are similarly suggestive, including dopamine–neurotensin binding associated with co-regulation of function, tricyclic antidepressants binding to penicillin and tetracycline-like antibiotics to enhance antibiotic activity, and the enhancement of peptide function when the peptide is combined with an antisense peptide to which it binds. In some of these cases, drug developers have already utilized the intersection of molecular complementarity and complementarity of function to develop methods of using the compounds together. In most cases they have not, leaving open many possibilities. 

## 4. Methods

### 4.1. Opioid–Adrenergic Compound Binding Test Methods

Opioids (methionine-enkephalin, morphine sulfate, methadone, and naloxone), various neurotransmitter and hormonal controls (serotonin, melatonin, histamine, and acetylcholine), and adrenergic compounds (epinephrine HCl, norepinephrine HCl, dopamine, l-3,4-dihydroxyphenylalanine (L-DOPA), propranolol, salbutamol, isoproterenol, tyramine, phenylephrine, octopamine, homovanillic acid, tyrosine, phenylalanine, and amphetamine), along with non-aminergic controls (glucose, ascorbic acid, and riboflavin), were obtained from Sigma-Aldrich (St. Louis, MO, USA). All opioids and opioid controls were tested for binding with all adrenergic compounds and non-aminergic controls. 

One compound was put in a pH 7.4 sodium phosphate buffer at 10 μM. The compounds to which this one compound might bind were dissolved in the same buffer at 1 mM and then serially diluted by thirds. We pipetted 100 μL aliquots of the first compound into the wells of a 96-well quartz crystal plate and added 100 μL of buffer solution. Similarly, all of the serial dilutions of the other compounds were pipetted into rows of wells in the crystal plate and 100 μL of buffer solution added to them. Finally, 100 μL of the first compound was combined with 100 μL of the serial dilutions of the other compounds. The absorbance of each compound alone and in combination was determined using ultraviolet spectroscopy from 190 nm to 350 nm in increments of 10 nm by means of a SpectraMax^®^ Plus automated scanning spectrophotometer with the SoftMax^®^ Pro program (Molecular Devices, San Jose, CA, USA). Beer’s Law holds that if two compounds do not interact, then the absorbance of the combination of the two compounds will be equal to the additive absorbance of each of the compounds on their own. If, however, there is binding, then the absorbance of the combination will differ in a concentration-dependent manner from the absorbance value predicted by adding the absorbances of the two, separate compounds. A binding constant can then be calculated by plotting the difference in absorbance as a function of the concentration of the varied compound and finding the inflection point of the resulting s-shaped curve. That inflection point is a close approximation to the Kd. All data were analyzed using Microsoft Excel. All combinations were performed at least twice. 

### 4.2. Opioid Peptide Binding Test Methods

After the solutions were made, a 96-well quartz crystal plate was prepared to be run through the spectrophotometer at room temperature (ca. 24 °C). The plate was set up to have the absorbance of each of the adrenergic compound dilutions measured on their own, and with each receptor peptide. The absorbance of each receptor peptide without the presence of the adrenergic compound was measured as well. The absorbance of each well was measured at every 10 nm increment from 190 nm to 260 nm. The maximum absorbance that can be measured was set to four. Each well had 200 μL of solution, so if the absorbance of one component was being measured, it was diluted by half with a phosphate buffer.

Spectrophotometry (SpectraMax ^®^ Plus scanning spectrophotometer with the SoftMax ^®^ Pro program) was used to measure the binding between opioid receptor peptides and opioid, adrenergic, and control compounds. Beer’s Law shows that if two compounds are not interacting in a solution, then the absorbance of that solution is equal to the additive absorbance of each of the compounds in a solution on their own. The binding between two compounds at a specific wavelength is found using the difference of the additive absorbances of each of the compounds in a solution on their own, and the absorbance found when they are in a solution together. If the measured absorbance is different than the additive absorbance of each compound, then that indicates some sort of molecular interaction. The binding can be quantified by graphing the difference in absorbance against the concentration of the compound varied to provide a binding curve. The data are analyzed by finding the additive absorbance and plotting the difference between this absorbance and the actual absorbance against the concentration of the adrenergic compound. The absorbance for the phosphate buffer is subtracted for each well before any calculations are done. All data were analyzed using Microsoft Excel, which reveals an “S” binding curve if binding is present. The binding constants were estimated from the half-saturation point.

### 4.3. Human Mu Opioid Receptor (muOPR) Expression and Purification

A purified human mu opioid receptor that was not reconstituted in cellular membranes was used for receptor binding experiments. The choice not to reconstitute the receptor into the membrane was due to the inability to find a cell membrane lacking adrenergic receptors. Since adrenergic receptors dimerize with opioid receptors (see Introduction), the presence of such receptors would have altered the results of the binding studies. In addition, adrenergic receptors obviously bind adrenergic agonists, which would have made it impossible to determine how much of an adrenergic ligand was binding to adrenergic receptors and how much to opioid receptors. And finally, there are multiple mechanisms of crosstalk between opioid and adrenergic receptors that we would not have been able to control for.

A codon-optimized human mu opioid receptor gene with an N-terminal deca-histidine tag in pQE-2 vector was used for protein expression in *E. coli* [72]. MuOPR expression was achieved using the muOPR transformed C43 (DE3) cell strain of *E. coli* in TB medium as reported earlier [72] with 0.4 mM IPTG induction at 18 °C for 24 h. Bacterial cell cultures were harvested by centrifugation at 4000× *g* for 20 min. The periplasmic fraction from the harvested cells was removed by osmatic shock [73]. Cells were resuspended in 7 mL of lysis buffer (20 mM Tris-HCl pH 8.0, 150 mM NaCl, 10% glycerol, 2 mM MgCl_2_, 10 μM E-64, 1 μM pepstatin-A, 10 μM leupeptin, 1 mM pefabloc SC, 2 mM β-mercaptoethanol, 1 mg/mL lysozyme, 30 U/mL DNAse) per gram of cell pellet and incubated on ice for 30 min with continuous stirring. The partial lysate was added with EDTA to a final concentration of 5 mM and passed through a high-pressure homogenizer, EmulsiFlex-C3, two to three times for efficient cell lysis. The lysate was clarified by centrifugation at 10,000× *g* for 40 min at 4 °C. The supernatant was collected to isolate the membrane fraction by centrifuging at 100,000× *g* for 1 h at 4 °C. The isolated membrane was solubilized in 20 mL of solubilization buffer (20 mM Tris–HCl pH 8, 300 mM NaCl, 10% Glycerol, 1% Fos-12, 10 μM E-64, 1 μM pepstatin-A, 10 μM leupeptin, 1 mM pefabloc SC) per gram of membrane for 3 h at 5 °C (cold room) with continuous stirring. The solubilized membrane sample was centrifuged at 100,000× *g* for 1 h at 4 °C and the supernatant was collected. Imidazole was added to the supernatant to a final concentration of 5 mM before starting purification using Ni-NTA resin in batch mode as reported [72]. Eluted fractions with muOPR were pooled, concentrated, and subjected to size exclusion chromatography (SEC) using HiLoad 16/600 Superdex 200 pg column equilibrated with 20 mM Tris pH 8, 150 mM NaCl, 0.1% Fos12, 10% Glycerol and 1 mM TCEP. SEC elution fractions were analyzed by SDS-PAGE and western blotting with a Monoclonal Anti-polyHistidine-Peroxidase antibody (Sigma Aldrich, St. Louis, MO, USA) for protein presence and purity before use. The estimated amount of muOPR in the pure sample was 309 μg in 3 mL.

### 4.4. Binding of Epinephrine and Opioids to muOPR Monitored by Ultraviolet Spectroscopy

A stock solution of 1200 μL of muOPR was formulated using 600 μL of muOPR (0.103 mg/mL) and 600 μL of 20 mM Tris buffer (pH 8), and 500 μL of 20 mM methionine-enkephalin (Sigma-Aldrich, St. Louis, MO, USA), morphine sulfate (Sigma-Aldrich), epinephrine HCl (Sigma-Aldrich), or other compounds tested, were freshly made with 20 mM Tris buffer and subjected to twelve serial dilutions by thirds in the buffer. We pipetted 100 μL of muOPR into twelve wells of a crystal 96-well plate and pipetted 100 μL of buffer into an additional twelve wells of the plate. Three muOPR and three buffer wells then received 10 μL of buffer, three received 5 μL of buffer plus 5 μL of epinephrine, three received 5 μL of buffer plus 5 μL opioid, and three received 5 μL opioid plus 5 μL epinephrine. The spectrum of the wells was then recorded from 190 nm to 260 nm using a SpectraMax ^®^ Plus scanning spectrophotometer using SoftMax ^®^ Pro software. The procedure described above was repeated an additional eleven times using compound dilutions of increasing concentration each time. 

The data were analyzed in Microsoft Excel. The raw spectra were processed by averaging the three runs of each condition and then subtracting the absorbance of the buffer alone at each volume. The triplicate data for each experimental condition were averaged. The compound + buffer data were subtracted from the muOPR + compound data at each volume to leave the spectrum of the muOPR under that experimental condition. The difference between the muOPR under that experimental condition and the muOPR merely diluted with the same volume of buffer was then calculated and this data was used to calculate the binding constant of the compound for muOPR. Because the final calculations involved several subtractions, error bars could be calculated.

## Figures and Tables

**Figure 1 ijms-20-04137-f001:**
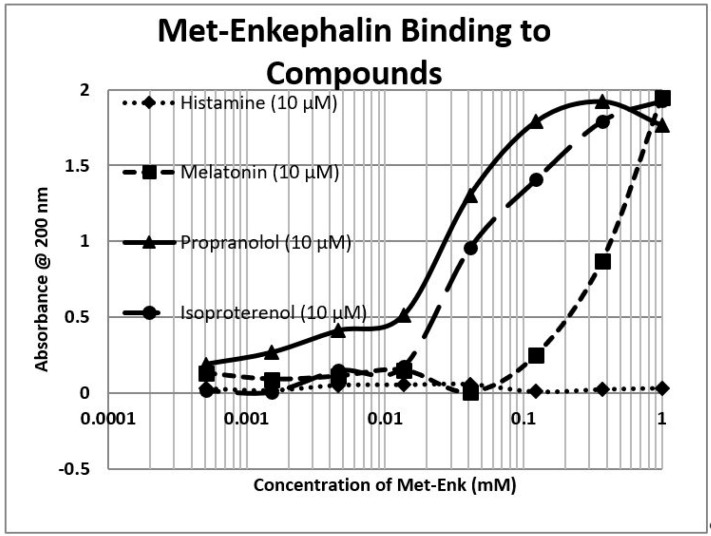
Ultraviolet spectroscopy study of methionine-enkephalin (Met-Enk) binding to various drugs and neurotransmitters. Note that propranolol appears to bind to Met-Enk in a two-phase manner, with high affinity binding around 3 μM and lower affinity binding around the 35 μM range. See Table 1 and Figure 2 and Figure 3 for additional data.

**Figure 2 ijms-20-04137-f002:**
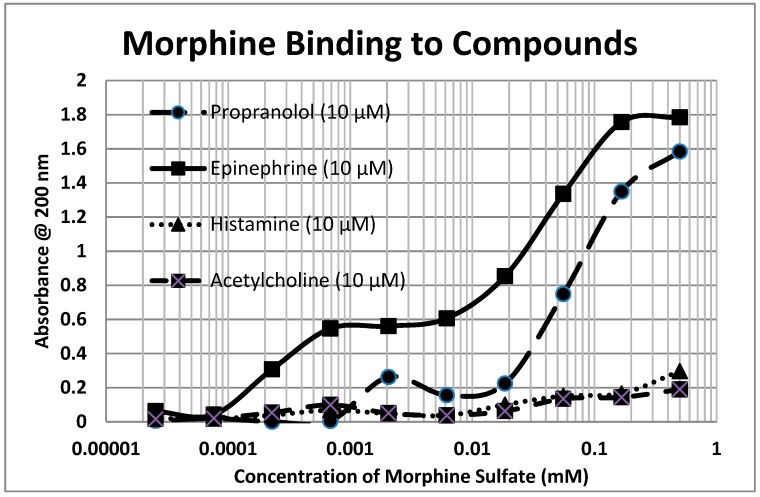
Ultraviolet spectroscopy study of morphine sulfate binding to various drugs and neurotransmitters. Note that epinephrine and propranolol bind to morphine in a two-phase manner, with high affinity binding in the high nanomolar to low micromolar range and lower affinity binding around the 40 μM to 50 μM range. See Table 1 and Figure 1 and Figure 3 for additional data.

**Figure 3 ijms-20-04137-f003:**
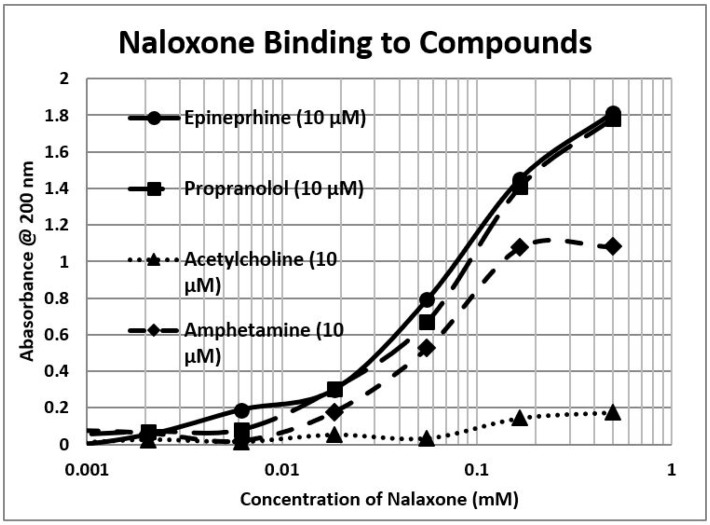
Ultraviolet spectroscopy study of naloxone binding to various adrenergic compounds. See Table 1 and Figure 1 and Figure 2 for additional data.

**Figure 4 ijms-20-04137-f004:**
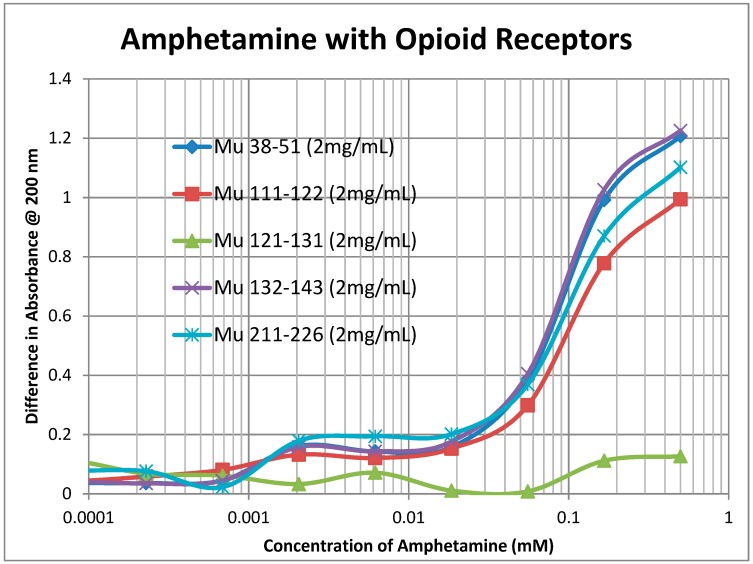
Binding of amphetamine to opioid receptor (muOPR) peptides. Note that there may be both high affinity binding to extracellular receptor peptides Mu 38, Mu 132, Mu 211, and possibly Mu 111, with Kd around 1.3 μM, and lower affinity binding at around 100 μM. No binding was observed between amphetamine and the transmembrane peptide Mu 121.

**Figure 5 ijms-20-04137-f005:**
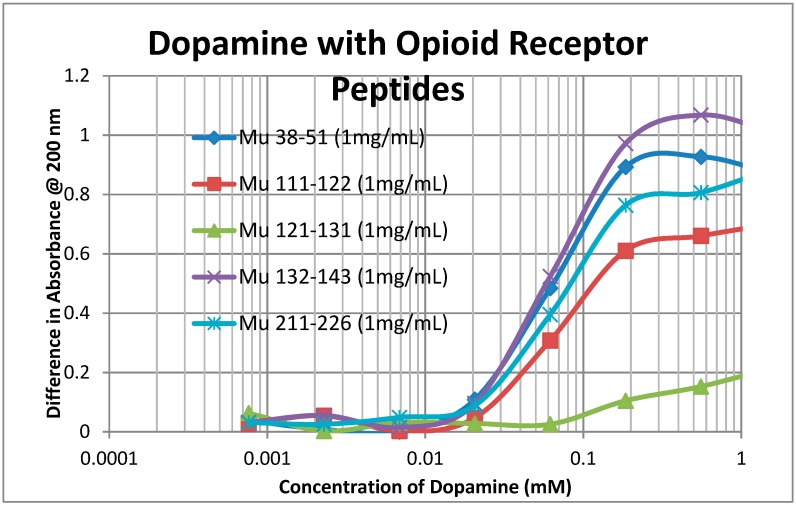
Binding of dopamine to mu opioid receptor (muOPR) peptides. Note that there is no high affinity binding evident as was observed with amphetamine (Figure 4). Dopamine displays only the lower affinity binding at around 70 μM to 90 μM. As with amphetamine (Figure 4), no measurable binding was observed between dopamine and the transmembrane peptide Mu 121.

**Figure 6 ijms-20-04137-f006:**
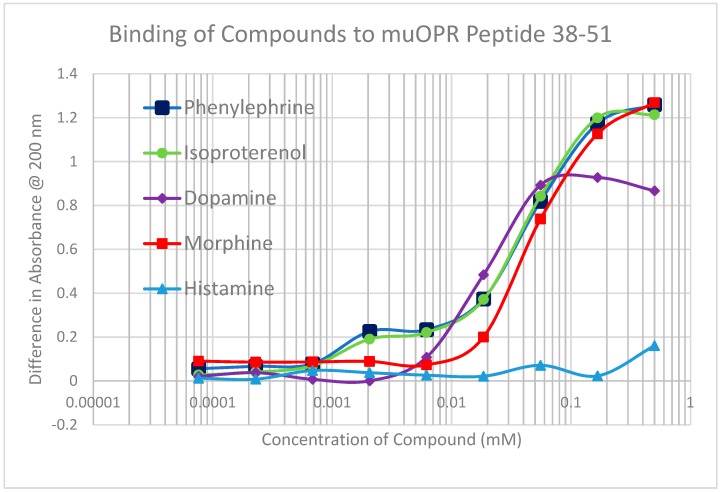
Binding of various bioactive compounds with the mu opioid receptor (muOPR) peptide 38–51. Note that there is high affinity binding at about 1.3 µM for phenylephrine and isoproterenol, similar to that observed with amphetamine (Figure 4). Morphine, like dopamine (Figure 5), displays only low affinity binding at about 60 μM, which is also shared by phenylephrine and isoproterenol. Histamine displayed no binding to Mu 38–51.

**Figure 7 ijms-20-04137-f007:**
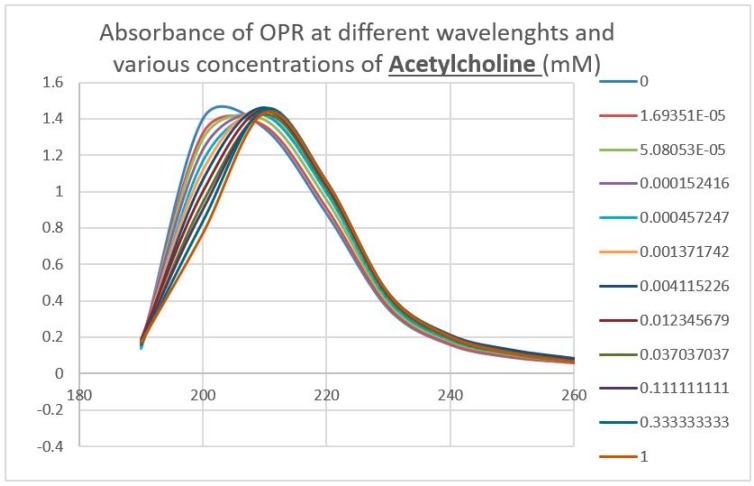
Effect of serial additions of acetylcholine at increasing concentrations on the UV spectrum of intact mu opioid receptors (muOPRs). The spectral shifts are due to the dilution of the receptor and do not differ significantly from those observed with equivalent additions of buffer without acetylcholine. Compare with Figure 8, Figure 9 and Figure 10.

**Figure 8 ijms-20-04137-f008:**
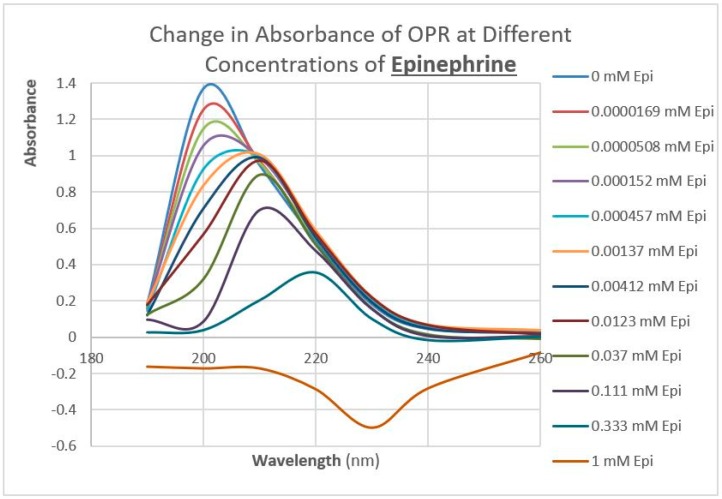
Effect of serial additions of epinephrine at increasing concentrations on the UV spectrum of intact mu opioid receptors (muOPRs). Compare with Figure 7.

**Figure 9 ijms-20-04137-f009:**
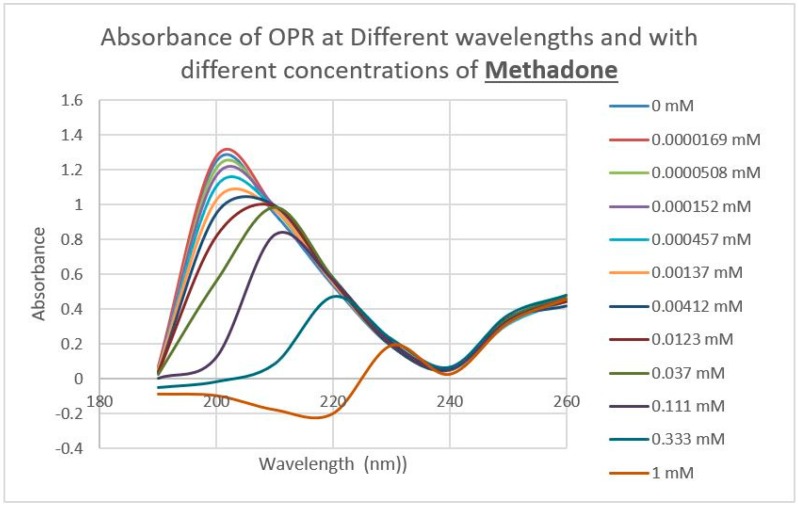
Effect of serial additions of methadone at increasing concentrations on the UV spectrum of intact mu opioid receptors (muOPRs). Compare with Figure 7 and Figure 8.

**Figure 10 ijms-20-04137-f010:**
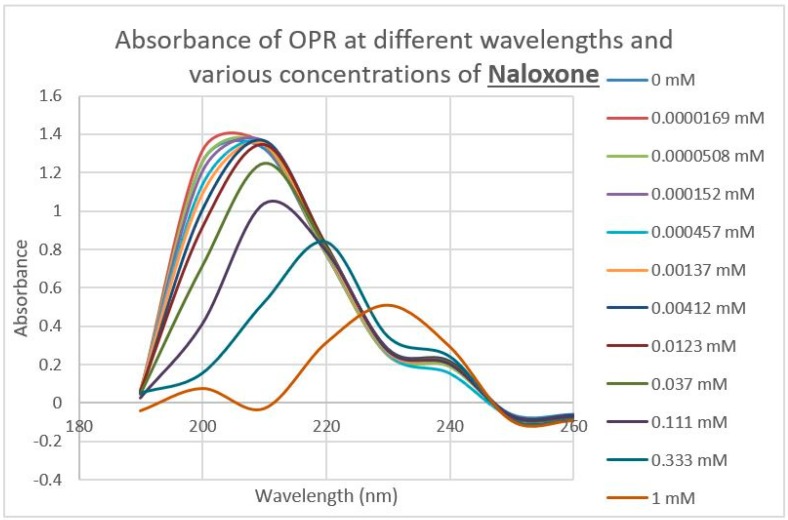
Effect of serial additions of naloxone at increasing concentrations on the UV spectrum of intact mu opioid receptors (muOPRs). Compare with Figure 7, Figure 8 and Figure 9.

**Figure 11 ijms-20-04137-f011:**
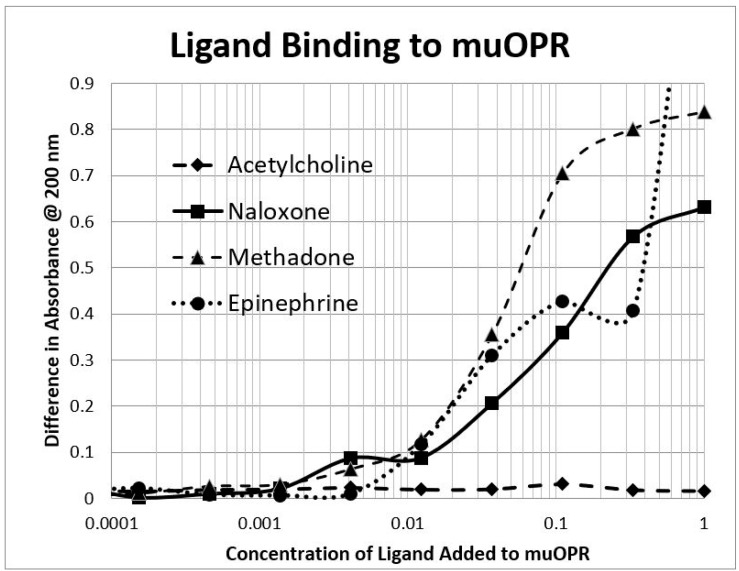
Ligand binding to intact mu opioid receptors (muOPRs). Acetylcholine demonstrated no observable bind while methadone, naloxone, and epinephrine bound at mid-micromolar concentrations.

**Figure 12 ijms-20-04137-f012:**
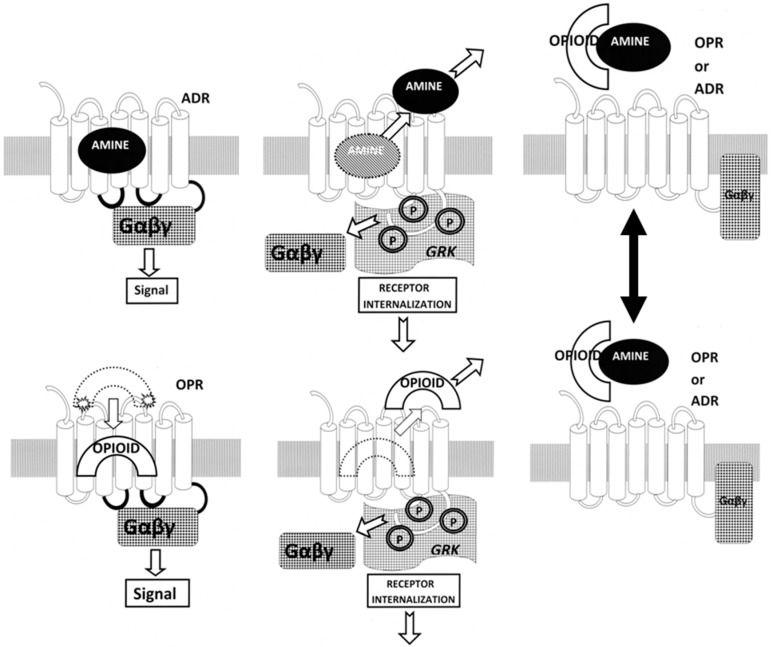
Schematic diagram of normal adrenergic receptor (ADR) and opioid receptor (OPR) function. Top Right: An adrenergic compound such as epinephrine binds to the ADR high-affinity site, initiating G-protein coupling (Gαβγ) to the intracellular loops of the receptor. Top Center: Binding is followed a short time later by the release of the ligand, phosphorylation (P) of the receptor by receptor kinases (GRK), and receptor inactivation and internalization. Bottom Right and Center: The same process characterizes opioid binding to OPR. Right Top and Bottom: Presence of both opioids and aminergic compounds activates both sets of receptors and initiates heterodimerization of the receptors. Notably, this receptor heterodimerization is mirrored by heterodimerization of the opioid and aminergic compounds as well, as demonstrated in this paper and in [78].

**Figure 13 ijms-20-04137-f013:**
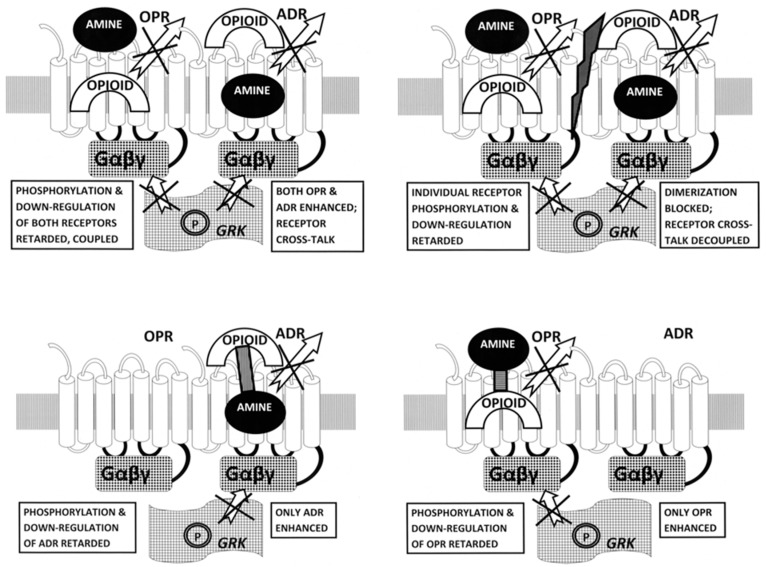
Drug development opportunities offered by a structural understanding of adrenergic receptor (ADR) co-evolution with opioid receptors (OPRs). Heterodimerization of ADR with OPR occurs when the ligands for both receptors are present (see Figure 10, Right) [8,9,10,11,12,13,14,15,16]. Top Left: The result of heterodimerization of ADR with OPR in the presence of both ligands is the enhancement of both receptors by the binding of the complementary compound to extracellular sites on the receptors. This binding of the complementary compounds enhances receptor activation by keeping the primary ligand in its high affinity pocket (either by allosteric changes in the structure of the receptors and/or by “capping” the binding site so that ligand is trapped for a longer period of time [40,56,64,65]. Heterodimerization also stimulates crosstalk between the receptors such that the G-protein coupling (Gαβγ) is enhanced and receptor kinase (GRK) binding is inhibited, so that phosphorylation of the receptors is prevented and down-regulation or internalization of the receptors is retarded (see Introduction). The overall effect is to increase the efficacy of any sub-maximal dose of the combined ligands, to increase their duration of activity, and to prevent tachyphylaxis and fade [40,56,64,65]. This model provides four novel opportunities for drug development. Top Left: The first opportunity is to optimize specific combinations of ADR and OPR agonists or antagonists to optimize receptor activation or deactivation. In particular, since opioid antagonists enhance ADR without activating OPR, and adrenergic antagonists enhance OPR without activating ADR, enhanced activation of particular sets of receptors is feasible. Top Right: Another drug development opportunity would be to develop drugs that inhibit heterodimerization (central slash), thereby decoupling the receptors, preventing crosstalk and possibly inhibiting the degree of co-activation of ADR with OPR. Bottom Left: A third opportunity would be to develop tethered drugs that optimize adrenergic ligand binding enhanced by a specific opioid or opioid antagonist. Again, since opioid antagonists enhance ADR without activating OPR, very specific activation (or deactivation, if an ADR antagonist were used) might be achieved. Bottom Right: Finally, using the same logic, it should be possible to develop tethered drugs that optimize opioid agonist or antagonist binding enhanced by a specific adrenergic agonist or antagonist.

**Table 1 ijms-20-04137-t001:** Results of ultraviolet spectroscopy study of binding between opioids, neurotransmitters, sugars, and vitamins. Binding constants are given in micromoles. Where two numbers are present, the experiment yielded a double curve, so that two binding constants were calculated: one represents high affinity binding, and the other, low affinity binding. M-Enk = methionine-enkephalin; Morph = morphine sulfate; Meth = methadone; NAL = naloxone; Amph = amphetamine; Prop = propranolol; Asc = ascorbic acid (vitamin C); Ribo = Riboflavin; l-DOPA = l-3,4-dihydroxyphenylalanine. * = binding confirmed by NMR spectroscopy [78]. # = binding confirmed by capillary electrophoresis [79].

Kd (µM) @ 200 nm	M-Enk	Morph	Meth	NAL	Amph	Prop	Asc	Glucose	Ribo
Epinephrine HCl	5.8/40 *	0.3/40 *	300	100	4.5	45/1000	90 ^#^	>1000	>1000
Norepinephrine HCl	5.3/35 *	0.4 *	70	80	80	>1000	110 ^#^	>1000	>1000
Dopamine	30 *	0.6 *	200	90	90	100	100	>1000	>1000
L-DOPA	70 *	>1000 *	80	55	11	15	100	>1000	>1000
Amphetamine	80 *	0.1 *	2.5	60		>1000	>1000	>1000	>1000
Propranolol	3.0/35	0.8/45	>1000	90	90		160	>1000	220
Salbutamol	30	0.3	>1000	180	0.6	1.0/130	160	>1000	50
Isoproterenol	40	0.1	13	13	0.1	11	130	>1000	>1000
Phenylephrine	30	0.13	>1000	20	17	20	>1000	>1000	>1000
Tyramine	12 *	50	210	90	60	80	200	>1000	40
Octopamine	80 *	3.2 *	>1000	55	100		220	>1000	30
Homovanillic Acid	80 *	>1000 *	>1000	85	1.5/90		>1000	>1000	70
Tyrosine	>1000	>1000	250	75	53	20	270	>1000	>1000
Phenylalanine	>1000	>1000	>1000	70	85		>1000	>1000	50
Serotonin	45 *	0.7 *	45	60	>1000	100	>1000	>1000	7
Melatonin	130	300	300	12	70	90	400	>1000	150
Histamine	>1000 *	>1000 *	>1000	210	70	110	>1000	>1000	>1000
Acetylcholine	80 *	>1000 *	>1000	>1000	>1000	>1000	>1000	>1000	>1000

**Table 2 ijms-20-04137-t002:** Binding constants (in micromoles) measured for various bioactive compounds interacting with the mu opioid receptor (muOPR) peptide, the dopamine D1 receptor (D1DR), the histamine type 1 receptor (H1HR), the beta 2 adrenergic receptor (B2AR), and control sequences from the insulin receptor (INSR). Where two values are presented separated by a slash, evidence of both high affinity and low affinity binding was observed, as illustrated in Figure 4 and Figure 6.

Kd (μM) @ 200 nm	Mor	Nalox	M-Enk	Epi	NorEpi	Amph	DOP	5HT	ACh	Hist
**MuOPR 38–51**	35	0.5/35	0.15/55	1.2/35	1.4/45	1.3/90	60	100	>1000	>1000
**MuOPR 111–122**	50	0.5/38	0.33/80	1.3/40	1.3/40	1.3/100	65	100	>1000	>1000
**MuOPR 121–131**	900	>1000	3.5/90	>1000	>1000	>1000	>1000	350	>1000	>1000
**MuOPR 132–143**	35	0.5/42	0.4 /70	1.4/35	1.4/40	1.1/85	60	100	>1000	>1000
**MuOPR 211–226**	30	1.0/45	1.0/65	1.2/40	1.3/45	1.2/90	65	90	>1000	>1000
**D1DR 89–98**	20	5	80	400	300	530	75	>1000	>1000	600
**D1DR 170–188**	310	150	150	900	>1000	230	300	>1000	>1000	>1000
**H1HR 77–87**	110	110	2.3/70	30	30	60	30	75	>1000	20
**B2AR 97–103**	1	6	130	120	600	130	70	120	>1000	>1000
**B2AR 175–188**	50	40	700	900	1000	2.3/600	800	>1000	>1000	>1000
**INSR 157–166**	60	200	100	>1000	140	400	>1000	110	>1000	110
**INSR 281–299**	>1000	>1000	>1000	200	>1000	>1000	>1000	900	>1000	>1000

**Table 3 ijms-20-04137-t003:** Binding constants (micromolar) for various ligands to intact mu opioid receptors (some data from [56]). Some binding curves displayed evidence of high and low affinity binding, indicated in the table by two or more Kd values separated by a slash. The presence of a question mark indicates that one of the curves is very small and possibly not reliable.

	Kd @ 210 nm (μM)	Kd @ 200 nm (μM)
Acetylcholine	>1000	>1000
Histamine	>1000	>1000
Ascorbic Acid (Vitamin C)	>1000	>1000
Naloxone	40	2.5/80
Methadone	50	60
Epinephrine	20/1000	30
Methionine-Enkephalin	10	0.8/15
Methionine-Enkephalin + Epinephrine	4	<0.01/4
Morphine	20	0.9/60
Morphine + Epinephrine	6	<0.01/0.9/9

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
