# Peer review of "Mutual Enhancement of Opioid and Adrenergic Receptors by Combinations of Opioids and Adrenergic Ligands Is Reflected in Molecular Complementarity of Ligands: Drug Development Possibilities"

_ijms, 2019, doi:10.3390/ijms20174137_

Round 1

Reviewer 1 Report

The paper of Root-Bernstein presents an interesting concept of opioid and adrenergic receptor co-evolution. The idea is interesting and significant, and I would really like to see it published in some time. However, the paper is not suitable for publication in the present form, and the extent of necessary modifications makes me suggest reconsidering it as a new submission after several improvements, rather than applying major revision.

The most important reservations concern the homology analysis. The Authors suggest that opioid and adrenergic receptors have a common ancestor - which is nothing new. They compare extracellular loops of these two groups and find that they are related - yes, they do! Both groups belong to the Rhodopsin family and obviously, these sequences are homologous. Authors ignore the fact, that ECL2 of both groups have completely different fold. Authors could check, for instance, the extent of ECL similarity between groups with analogical evolutionary distance, and if it would be significantly larger in opioid-adrenergic pair, such results would support their conclusions, while analyses presented in the paper do not. Notably, Authors state that they found homology between enkephalins and opioid receptors 'within the opioid ligand binding site of the opioid receptors' (line 297), while figure12, which was supposed to support this statement, clearly shows that the region of interest is located in the C-terminal tail (most cases) or ECL2/TM4 of the receptor, and only the latter could contribute slightly to binding of large peptides. In the 6ddf Cryo-EM structure none of these regions contribute to binding of DAMGO.

In their discussion on evolutionary relationships, Authors completely ignore the work of Pele et al., which have shown that opioid receptors, together with chemokine, somatostatin and purinergic receptors are separated from the rest of the rhodopsin family branches by a deletion in the TM2, and therefore these families should be considered together in the evolutionary tree of GPCRs. While opioid receptors indeed have an aspartic acid in the 3.32 position, which resembles aminergic receptors, other members of the Pele's G1 family do not have this feature, and this fact should be considered and discussed.

There are also some remarks on in vitro methodology. Authors obtain the opioid receptor in bacteria. The description suggests, that receptor are not purified and reconstituted into eukaryotic membranes, but they stay in the bacterial membranes. This can greatly affect results and should be validated, especially since more reliable methods are easily available. Authors could, for instance, use cAMP accumulation assay on eukaryotic cells, which is not very expensive (division arrested cells are relatively inexpensive) and would be more confident. Optionally, eukaryotic membrane preparations with GPCRs embedded are commercially available; in such case, binding could be measured e.g. with ITC. These methods would be also much more reliable than UV spectroscopy in this case. I know about some applications of this method to enzymes, but I'm not really convinced if it would also apply to membrane preparations. Authors should cite appropriate research or validate the method on their own, and provide results. Now, I find it confusing that with this method, Kd of morphine or methadone binding to mu opioid receptor was found to be in a micromolar scale (table without number or caption, located between lines 269 and 270).

There are also some minor remarks considering style (long sentences, like lines 71-73) ald slightly chaotic presentation of the results.

The most interesting and reliable results are presented in Tables 1 and 2. They suggest that the conclusions drawn by Authors could be correct. However, it's definitely not enough and further data are needed.

Author Response

The paper of Root-Bernstein presents an interesting concept of opioid and adrenergic receptor co-evolution. The idea is interesting and significant, and I would really like to see it published in some time. However, the paper is not suitable for publication in the present form, and the extent of necessary modifications makes me suggest reconsidering it as a new submission after several improvements, rather than applying major revision.

The most important reservations concern the homology analysis. The Authors suggest that opioid and adrenergic receptors have a common ancestor - which is nothing new. They compare extracellular loops of these two groups and find that they are related - yes, they do! Both groups belong to the Rhodopsin family and obviously, these sequences are homologous. Authors ignore the fact, that ECL2 of both groups have completely different fold. Authors could check, for instance, the extent of ECL similarity between groups with analogical evolutionary distance, and if it would be significantly larger in opioid-adrenergic pair, such results would support their conclusions, while analyses presented in the paper do not. Notably, Authors state that they found homology between enkephalins and opioid receptors 'within the opioid ligand binding site of the opioid receptors' (line 297), while figure12, which was supposed to support this statement, clearly shows that the region of interest is located in the C-terminal tail (most cases) or ECL2/TM4 of the receptor, and only the latter could contribute slightly to binding of large peptides. In the 6ddf Cryo-EM structure none of these regions contribute to binding of DAMGO.

In their discussion on evolutionary relationships, Authors completely ignore the work of Pele et al., which have shown that opioid receptors, together with chemokine, somatostatin and purinergic receptors are separated from the rest of the rhodopsin family branches by a deletion in the TM2, and therefore these families should be considered together in the evolutionary tree of GPCRs. While opioid receptors indeed have an aspartic acid in the 3.32 position, which resembles aminergic receptors, other members of the Pele's G1 family do not have this feature, and this fact should be considered and discussed.

Given that all three reviewers had some sort of difficulty with the evolutionary analysis, but found the experimental part of the paper valuable, we have decided to remove the evolutionary analysis completely and will explore it in a separate manuscript.

There are also some remarks on in vitro methodology. Authors obtain the opioid receptor in bacteria. The description suggests, that receptor are not purified and reconstituted into eukaryotic membranes, but they stay in the bacterial membranes. This can greatly affect results and should be validated, especially since more reliable methods are easily available. Authors could, for instance, use cAMP accumulation assay on eukaryotic cells, which is not very expensive (division arrested cells are relatively inexpensive) and would be more confident. Optionally, eukaryotic membrane preparations with GPCRs embedded are commercially available; in such case, binding could be measured e.g. with ITC. These methods would be also much more reliable than UV spectroscopy in this case. I know about some applications of this method to enzymes, but I'm not really convinced if it would also apply to membrane preparations. Authors should cite appropriate research or validate the method on their own, and provide results. Now, I find it confusing that with this method, Kd of morphine or methadone binding to mu opioid receptor was found to be in a micromolar scale (table without number or caption, located between lines 269 and 270).

We understand the Reviewer’s concerns but chose the binding method we used for because all eukaryotic membranes contain aminergic receptors. These dimerize with opioid receptors to alter their affinity for opioid ligands. In addition, as we have demonstrated in this and other papers, opioid ligands bind directly to aminergic receptors, again complicating analysis of opioid receptor binding. Similarly, aminergic ligands bind directly to aminergic receptors masking their affinity for opioid receptors. And finally, there are many mechanisms of opioid-adrenergic receptor cross-talk that we could not reasonably control for (these are discussed in the Introduction to the paper). In short, the only way to be certain that what we measured was opioid and/or aminergic binding solely to opioid receptors was to purify them without reconstituting them in eukaryotic membranes. This method obviously decreases binding affinity since the receptors are no longer in their preferred conformations, which is the trade-off of using this method. We have now added this explanation both in the text (Results and in Methods) and we have pointed out the limitations of the method in the Results section.

There are also some minor remarks considering style (long sentences, like lines 71-73) ald slightly chaotic presentation of the results.

The manuscript has been rewritten extensively to shorten and clarify the presentation.

The most interesting and reliable results are presented in Tables 1 and 2. They suggest that the conclusions drawn by Authors could be correct. However, it's definitely not enough and further data are needed.

The Reviewer has not been specific about what further data would be relevant so we are not sure how to respond to this point or to address it with further experiments.

Reviewer 2 Report

I found this manuscript's topic to be quite interesting , however I have serious issues with the subject material and the conclusions made from the limited data provided.  I found the manuscript to be overly long, very difficult to read, and somewhat disjointed and rambling with a tendency to quickly change focus.  I also find that many aspects of the work are highly speculative and I do not believe that the speculative nature of the data firmly supports the conclusions made by the authors.  This article is to me less of a typical scientific publication of specific experimental data and more of a perspective -type of publication, serving a very different purpose.  In the end, I did like the work and was intrigued by the premise, but I just didn't find enough specific data to firm up the paper. I also disagree with the authors conclusions about how a study like this might inform the drug discovery process, I'd like to see a much better discussion of this point with some specific examples cited.  In general,  I would recommend making the paper much more concise (it is too long and is very difficult to read and follow)and readable, and perhaps adding more specific data to try to firm up the conclusions.

Author Response

I found this manuscript's topic to be quite interesting , however I have serious issues with the subject material and the conclusions made from the limited data provided.  I found the manuscript to be overly long, very difficult to read, and somewhat disjointed and rambling with a tendency to quickly change focus.  I also find that many aspects of the work are highly speculative and I do not believe that the speculative nature of the data firmly supports the conclusions made by the authors.  This article is to me less of a typical scientific publication of specific experimental data and more of a perspective -type of publication, serving a very different purpose.  In the end, I did like the work and was intrigued by the premise, but I just didn't find enough specific data to firm up the paper. I also disagree with the authors conclusions about how a study like this might inform the drug discovery process, I'd like to see a much better discussion of this point with some specific examples cited.  In general,  I would recommend making the paper much more concise (it is too long and is very difficult to read and follow)and readable, and perhaps adding more specific data to try to firm up the conclusions.

As noted in our response to Reviewer 1, we have eliminated the evolutionary aspect of the manuscript since it is the most theoretical. We have tightened up and shortened the rest of the manuscript. And we have strengthened the discussion the drug development implications by adding a number of examples of using complementary ligands to modulate receptor activity (in Discussion section).  We also note that Reviewer 3 appears to disagree with this Reviewer concerning the drug development implications….

Reviewer 3 Report

This is a well written manuscript detailing evidence regarding the co-evolution of the opioid and adrenergic receptors. The manuscript details a set of binding experiments between opioid and adrenergic compounds. The authors then evaluate binding of adrenergic compounds to opioid receptors. The authors additionally study the sequence similarity/conservation between the receptor sequences. The authors them postulate around the evolutionary biology of the receptors and how this can be taken advantage of in the context of drug discovery/development. 

It is suggested that the authors provide additional or clarity information on the following prior to publication:

·     The authors should discuss if the evolution of these receptor classes is unexpected based on the well-known interactions between opioid and adrenergic compounds.

·     The choice of control compounds such as histamine, acetylcholine, glucose, etc. should be provided.

·     From a drug development perspective, understanding the binding to the receptors and cross-reactivity is of high interest, no matter the evolution of the receptors. The authors may want to add language to decouple the determination of the evolution of the receptors from the drug development interest.

Author Response

This is a well written manuscript detailing evidence regarding the co-evolution of the opioid and adrenergic receptors. The manuscript details a set of binding experiments between opioid and adrenergic compounds. The authors then evaluate binding of adrenergic compounds to opioid receptors. The authors additionally study the sequence similarity/conservation between the receptor sequences. The authors them postulate around the evolutionary biology of the receptors and how this can be taken advantage of in the context of drug discovery/development.

It is suggested that the authors provide additional or clarity information on the following prior to publication:

·     The authors should discuss if the evolution of these receptor classes is unexpected based on the well-known interactions between opioid and adrenergic compounds.

As noted above in response to Reviewers 1 and 2, we have opted to remove the discussion of the co-evolution of these receptor classes so as to streamline the paper and focus it on experimental results. This also addresses Reviewer 3’s suggestion (below) that we decouple the evolutionary aspect of the problem from the drug development issue.

·     The choice of control compounds such as histamine, acetylcholine, glucose, etc. should be provided.

·     From a drug development perspective, understanding the binding to the receptors and cross-reactivity is of high interest, no matter the evolution of the receptors. The authors may want to add language to decouple the determination of the evolution of the receptors from the drug development interest.

As noted above, we have now eliminated the evolutionary discussion, decoupling it from the drug development interest.

Round 2

Reviewer 1 Report

The Authors did not address the issues pointed in my previous review sufficiently. For instance, they stated that:

"The most interesting and reliable results are presented in Tables 1 and 2. They suggest that the conclusions drawn by Authors could be correct. However, it's definitely not enough and further data are needed.

The Reviewer has not been specific about what further data would be relevant so we are not sure how to respond to this point or to address it with further experiments."

...while a couple of lines above there was a sentence:

"Authors could, for instance, use cAMP accumulation assay on eukaryotic cells, which is not very expensive (division arrested cells are relatively inexpensive) and would be more confident. Optionally, eukaryotic membrane preparations with GPCRs embedded are commercially available; in such case, binding could be measured e.g. with ITC. These methods would be also much more reliable than UV spectroscopy in this case."

I still think that additional validation is needed, since the environment of investigated receptors is far from native. Moreover, it was not checked if presence of the membrane fragments does not affect UV-Vis measurements - this should be validated with another method, or appropriate source should be cited.

In opinion of the Authors, it is better to use bacterial membranes with specific bacterial lipids, lacking cholesterol, than to use eukaryotic membranes, and that there is no other option to examine opioid receptors without interference with other GPCRs. Therefore, Authors ignore the possibilities that can be found in literature for some time, which are successfully used for many GPCRs, like SMALPs or nanodiscs (e.g. Dörr et al. European biophysics journal, 2016). For instance, Authors could obtain receptors from bacterial source, purify them and reconstitute into artificial eukaryotic-like nanodiscs. One should be aware, that allosteric factors strongly affect GPCR function, e.g. LY2033298 makes mAchR sense choline as its agonist. There are also studies indicating that GPCRs migrate to areas of particular membrane composition upon activation, showing that membrane-receptor interactions are important for activation. Effect of bacterial membrane on the structure of investigated GPCRs is hard to predict and therefore makes such experiments difficult to evaluate.

It is possible that the method used by the Authors is appropriate, but it should be validated, as already mentioned, e.g. by ITC measurements, and analysis of membrane influence on function of investigated receptors.

Author Response

We thank the Reviewer for clarification.  We did not understand that what was requested was validation of the UV binding method using another method since UV binding is so well-established. We, and other groups over the past 30 years, have validated the results from UV binding studies using numerous other methods including NMR spectroscopy, IR spectroscopy, circular dichroism studies, etc. We are not aware of any head-to-head ITC-UV studies, but since all colligative properties yield very similar results, there is little reason to doubt the validity of the measurements.

More importantly, we have now added references in lined 217-218 and 237-259 concerning studies of the binding of some adrenergic compounds to opioid receptors by other groups using other methods. We believe that these published reports serve the same purpose as the Reviewer's request that we perform validation experiments.

The paper has been spell-checked and revised to eliminate typos and grammatical issues.

Reviewer 2 Report

The authors have made a solid attempt to shorten and clarify the manuscript, and have done a good job of editing out the more speculative/controversial aspects, and have chosen to expand and focus on the potential drug development implications of their work. I found the revised manuscript much easier to read and understand, and much less random and disjointed.  The manuscript is now in a more acceptable form I believe for publication in the journal.  There are still a few minor punctuation and grammatical errors that can be easily fixed(requires minor editing), but in terms of the content I think the authors have done a good job of revising the manuscript.  I now find the revised manuscript to be acceptable for publication in the journal once the few simple grammar and punctuation errors are corrected .  I appreciate the efforts of the authors to make this a more concise, readable, and solid manuscript.

Author Response

We thank the Reviewer for the positive feedback and have reread the manuscript to correct grammatical errors and submitted it to a detailed spell-check. We hope that this process has caught all of the remaining errors.